# Comparative Proteomic Identification of Ram Sperm before and after In Vitro Capacitation

**DOI:** 10.3390/ani14162363

**Published:** 2024-08-15

**Authors:** Ya-Le Chen, Chun-Yan Li, Peng-Hui Wang, Ru Wang, Xian Zhuo, Yan Zhang, Shi-Jia Wang, Zhi-Peng Sun, Jia-Hong Chen, Xiao Cheng, Zi-Jun Zhang, Chun-Huan Ren, Qiang-Jun Wang

**Affiliations:** 1College of Animal Science and Technology, Anhui Agricultural University, Hefei 230036, China; chenyale1@163.com (Y.-L.C.); wangpenghui15987@163.com (P.-H.W.); wangru3127@163.com (R.W.); zzhuoxian@163.com (X.Z.); wshijia2024@163.com (S.-J.W.); fsunzhipeng@163.com (Z.-P.S.); chenjiahong@ahau.edu.cn (J.-H.C.); chengxiao@ahau.edu.cn (X.C.); zhangzijun@ahau.edu.cn (Z.-J.Z.); 2Yunnan Animal Science and Veterinary Institute, Kunming 650224, China; chunyanli2023@163.com (C.-Y.L.); haijippp@163.com (Y.Z.); 3Center of Agriculture Technology Cooperation and Promotion of Dingyuan County, Chuzhou 233200, China

**Keywords:** spermatozoa, capacitation, tandem mass tag, proteomic, bioinformatics

## Abstract

**Simple Summary:**

Capacitation confers competency to spermatozoa to fertilize the oocyte, yet its regulatory mechanisms are not fully understood. Thus, we aimed to investigate the comparative proteomic profiling in ram spermatozoa under non-capacitating (NC) and capacitating (CAP) conditions in vitro using a liquid chromatography–tandem mass spectrometry (LCMS/MS) combined with tandem mass tag (TMT) labeling strategy. Functional enrichment analysis indicated that the differentially abundant proteins Prune Exopolyphosphatase 1, Galactose-1-Phosphate Uridylyltransferase, and ATP Citrate Lyase were strictly related to energy production and conversion, and Phosphoglycolate phosphatase, Glucosamine-6-Phosphate Deaminase 1 and 2 were related to metabolism, RNA processing, and vesicular transport pathways. Furthermore, the networks of protein–protein interaction indicated a strong interaction among these differential proteins in annotated pathways such as ubiquitin and transport metabolism. Together, our results provided the database for studying the differentially expressed proteins during ram sperm capacitation.

**Abstract:**

Ram sperm undergo a sequence of physiological and biochemical changes collectively termed as capacitation to perform oocyte fertilization. However, the protein changes induced by capacitation remain in need of further exploration. Thus, the present study investigated the comparative proteomic profiling in ram spermatozoa under non-capacitating (NC) and capacitating (CAP) conditions in vitro using a liquid chromatography–tandem mass spectrometry combined with tandem mass tag labeling strategy. As a results, 2050 proteins were identified and quantified; 348 of them were differentially abundant, with 280 of the proteins upregulated and 68 of the proteins downregulated between the CAP and NC spermatozoa, respectively. Functional enrichment analysis indicated that the differentially abundant proteins Prune Exopolyphosphatase 1, Galactose-1-Phosphate Uridylyltransferase, and ATP Citrate Lyase were strictly related to energy production and conversion, and Phosphoglycolate phosphatase, Glucosamine-6-Phosphate Deaminase 1 and 2 were related to metabolism, RNA processing, and vesicular transport pathways. Furthermore, the networks of protein–protein interaction indicated a strong interaction among these differential proteins in annotated pathways such as ubiquitin and transport metabolism. Our findings indicate that capacitation progress might be regulated through different pathways, providing insights into mechanisms involved in ram sperm capacitation and fertility.

## 1. Introduction

Mammalian spermatozoa cannot fertilize oocytes immediately after ejaculation; they need to acquire fertilizing capacity in the female genital tract through a time-dependent process called capacitation [1,2]. However, as a highly specialized cell, sperm is thought to be largely quiescent in terms of transcriptional and translational activity [3,4]. In recent years, some scholars have suggested that translation of new proteins from mRNA transcripts can take place during sperm capacitation [5]. Sperm capacitation is mainly dependent on post-translational modifications, such as phosphorylation of pre-existing sperm proteins, and phosphorylation of tyrosine residues is a consistent indicator of sperm capacitation [6]. Many studies have shown that inhibiting tyrosine phosphorylation can block sperm capacitation, acrosome reactions, and in vitro fertilization [7,8]. Studies have identified phosphorylated and other proteins that play key roles in sperm capacitation in humans, mice, wild boars, buffalo, and other species [8,9,10,11,12,13]. However, the proteomic characteristics of sheep sperm before and after capacitation have not been studied until now.

Proteomic studies have provided novel information for sperm capacitation. There are a cascade of proteins involved in energy production, signaling, cell–cell adhesion, motility performance, changes in ROS and pHi, cytoskeleton reorganization, and ionic transport during sperm capacitation [14,15], in humans [16,17,18], mice [12,19], boars [20] and other mammals; examples are NADPH oxidase 5 (NOX5) as an active generator of ROS that expresses in human and ram sperm [21], and Proacrosin as a carbohydrate-binding protein that is associated with several low-molecular-weight proteins that may be related to active forms of acrosin in human, bull, boar and ram spermatozoa. More specifically, elevated ROS levels increase the levels of tyrosine-nitrated and S-glutathionylated proteins in human sperm capacitation and AR [22,23,24,25]. In ram spermatozoa, dihydrolipoamide acetyltransferase, a component of pyruvate dehydrogenase complex (DLAT), is upregulated after capacitation; this is associated with energy production, and also affects ROS levels during capacitation [26]. In addition, studies have demonstrated that Ras-related C3 botulinum toxin substrate 1 protein (Rac1), as a member of Rho protein family, has been found to participate in the actin cytoskeleton remodeling of the acrosome apical region during sperm capacitation [27,28]. Ionic channel proteins such as Anoctamin 1 (TMEM16A), and Transient Receptor Potential Vanilloid family member sub type 1 and 4 (TRPV1 and TRPV4) activate ion currents and affect hyperactivated motility during capacitation [29,30,31]. Furthermore, the following are also involved: tyrosine phosphorylation-related proteins, such as A-kinase anchoring proteins (AKAPs) and Sperm plasma membrane-associated protein (SMAP) in humans [32,33], Phosphorylated protein of 32 KD (p32) in boar [34], Ca2+-binding tyrosine-phosphorylation-regulated protein (CABYR) and Serpin peptidase inhibitor clade E, member 2 (SERPINE2) in mice [35,36]. All this knowledges about sperm capacitation has been obtained based on proteomic techniques.

Because the quality of semen has a great impact on the fertility of sheep, the obstacle of the sperm fertilization process is an important factor for the low sperm fertilization rate, and sperm capacitation is a prerequisite for successful sperm fertilization. In the current study, we employed a quantitative proteomic technique to investigate changes in protein components during ram spermatozoa capacitation in vitro. This may help to identify the differential proteins during capacitation that are essential for maintaining sperm function, aiming to replenish knowledge of the molecular mechanism leading to post-ejaculation ram sperm maturation from a proteomic perspective.

## 2. Materials and Methods

### 2.1. Experimental Design and Workflow

This experimental research involving animals was carried out following welfare guidelines, and ethical approval was provided by the Animal Care Committee, Anhui Agricultural University (No. AHAU2020023, 4 January 2020). The TMT-based quantitative proteomic technique was used to identify and characterize the protein differences in spermatozoa before and after in vitro capacitation. The experimental design and workflow are shown in Appendix A. Each three out of the nine samples were mixed randomly in equal volumes to form a biosample, with a total of three mixed biosamples in each group (non-capacitating group and capacitating group). The protein extraction, TMT labeling, and mass spectrometry were performed for protein identification and were followed by bioinformatics analysis.

### 2.2. Animal and Sample Collection

Healthy adult rams (n = 9) with no significant differences in age (2 years of age), weight, height, and body length were selected. They were housed at a breed protection farm of the Jianghuai Watershed Comprehensive Experimental Station (Dingyuan, China). Fresh ejaculates were collected via an artificial vagina from nine rams in the presence of a teaser ewe. After collection, sperm motility and density were immediately evaluated using the Computer-Assisted Sperm Analysis System (CASA, sperm classanalyzer-5.4.0.0; MICROPTIC Supply, Barcelona, Spain). Only those ejaculates with a motility greater than 80% and a density greater than 3 × 10^9^/mL were used in the subsequent experiment [37]. To ensure the amount of qualified ejaculates, the ram could be collected again two days later. Three out of the nine samples were mixed randomly in equal volumes to form a biosample, and a total of three mixed biosamples were used in our experimental replication (n = 3). The above three biosamples were immediately prediluted with the sperm TALP culture medium (IVL03-100ML, Caisson, Smithfield, VA, USA) and centrifuged at 700× *g* for 10 min to isolated sperm (deposit) and seminal plasma (liquid supernatant) [38]. Then, sperm were wash twice with PBS by centrifugation (400× *g*, 5 min) and divided into two parts: one part was used as the non-capacitating (NC) sample resource, the other part was used for capacitation.

### 2.3. Sperm Capacitation

For sperm capacitation, each deposit was washed with 1 mL sperm–TALP culture medium and centrifuged at 400× *g* for 5 min. Afterwards, each deposit was rediluted in the sperm–TALP culture medium to 1 × 10^7^ sperm/mL. Every 100 µL was carefully transferred into the bottom of round-bottom tube, which contained 1 mL of pre-heated sperm–TALP culture medium, 50 µg sodium heparin and 2% (*v*/*v*) estrous ovine serum, collectively referred to as capacitation solution. The round-bottom tube was placed gently at an angle of 45°, followed by sperm capacitation using the upstream method at 38.5 °C and 5% (*v*/*v*) CO_2_ for 240 min to obtain the capacitating (CAP) sample resource [26,39]. To stimulate AR, 10 µL of the calcium ionophore A23187 (1.0 mM in DMSO, diluted to 100 µM in Sp-TALP when used, CSN pharm, Chicago, IL, USA) was added to 90 µL of the spermatozoa from the above NC and CAP groups to achieve the final concentration of 10 µM, before incubation at 38.5 °C for 30 min [40]. After incubation, all samples from two groups were centrifuged (500× *g*, 5 min) to obtain the deposit (sperm), the sperm were wash thrice with PBS by centrifugation (400× *g*, 5 min), and aliquots of 1 × 10^7^ AR sperm were collected. A small part of each sample (10 µL) was reserved for detection of sperm capacitation, and the rest was stored immediately at −80 °C until TMT and parallel reaction monitoring (PRM) analyses.

For determining the capacitation status, the CAP group and the CN group were fixed in 100 µL of PBS with 2% (*v*/*v*) paraformaldehyde for 10 min at 24 °C, and centrifuged at 200× *g* for 5 min. The precipitate was washed twice with 1.5 mL ammonium acetate (0.1 moL/L) and the precipitate was resuspended in 200 µL PBS. The suspension liquid (20 µL) was pipetted into each tube for smearing and stained with 0.25% Coomassie Brilliant Blue G-250 for 5 min. Samples were air-dried and observed under a 200-times (10 × 20) phase-contrast optical microscope combined with an OPLENIC imaging system for the determination of sperm capacitation [8,41]. Additionally, to support these capacitive changes in sperm, we performed Western blot analysis of the level of tyrosine phosphorylation protein (Tyr-P), which is a known reliable sign of sperm capacitation. In Western blot, anti-Tyr-P (CST9411, 1:20, Ipswich, MA, USA) was used, and anti-α-tubulin (Proteintech66031-1-Ig, 1:300, Wuhan, China) was used as a control [40].

### 2.4. Protein Extraction and Digestion

Each sample (approximately 1 × 10^7^ sperm cells) was sonicated three times on ice, using a high-intensity ultrasonic processor (Scientz) in lysis buffer 8 moL/L urea (Sigma, St. Louis, MO, USA), 2 mmoL/L EDTA (Sigma, St. Louis, MO, USA), 10 mmoL/L DDT (Sigma, St. Louis, MO, USA) and 1% protease inhibitor cocktail (SparkJade, Shandong, China), followed by centrifugation at 20,000× *g* for 10 min at 4 °C [42]. Then, the supernatant was transferred into a clean tube for the protein extraction and digestion, according to a previous report [43]. The protein concentration was determined by a BCA Kit (Beyotime, Shanghai, China), detected by 10% (*w*/*v*) SDS-PAGE, and 20 µg protein was loaded onto each electrophoretic lane. Six electrophoretic bands were clear and uniform, with good parallelism in each gel swim-lane (Appendix A), which indicated that proteins were good for subsequent experiments. Based on the BCA results, each protein sample (30 mg) was subjected to enzymolysis.

### 2.5. TMT Labeling

After trypsin digestion, the peptides were desalted using a Strata X C18 SPE column (Phenomenex, Torrance, CA, USA) and dried by vacuum centrifugation. The desalted peptides (100 µg) from each sample were dissolved in 0.5 moL/L TEAB solution and labeled using the TMT Labeling Kit (ThermoFisher Scientific, Waltham, MA, USA) according to the manufacturer’s protocol. In brief, the above peptide dissolved solution was incubated with the TMT regent (1 unit of labeling reagent was used for 100 µg of peptide), which was reconstituted in 24 μL ACN, for 2 h at room temperature. Then, three pooled fractions of the capacitating sperm group were labeled with 126 (CAP1), 127 (CAP2), and 128 (CAP3) tags, while the non-capacitating sperm group was labeled with 129 (NC1), 130 (NC2), and 131 (NC3) tags, respectively. The reaction was stopped with 8% ammonium hydroxide. Differently labeled peptides were mixed equally, desalted and vacuum dried [42,43,44].

### 2.6. High-pH Liquid Chromatographic (HPLC) Separation of Peptides

Peptides were fractionated by high-pH reverse-phase HPLC, using Betasil C18 column (5 μm particles, 10 mm ID, 250 mm length, Thermo, Waltham, MA, USA). Briefly, peptides were first separated with a gradient of 8% to 32% acetonitrile (pH 9.0) over 60 min, into 60 fractions. Following this, the peptides were combined into 14 fractions and dried by vacuum centrifuging.

### 2.7. LC-MS/MS Analysis

The peptides were dissolved in 0.1% (*v*/*v*) formic acid (solvent A) and directly loaded onto a reversed-phase C18 analytical column (Thermo Scientific, 15 cm length, 75 μm i.d.). The gradient comprised an increase in solvent B from 8% to 23% (0.1% formic acid in 90% acetonitrile) over 26 min, 23% to 35% for 8 min, climbing to 80% for 40 min, and sustained at 80% for the last 3 min, all at a constant flow rate (500 nL/min) on an EASY-nLC 1000 UPLC system.

The separated peptides were injected into nano electrospray ionization (NSI) sources for ionization, with the voltage set to 2.0 kV, then tandem mass spectrometry (MS/MS) was performed with Q ExactiveTM Plus (Thermo Scientific, USA), which was coupled online to the UPLC. Intact peptides were detected in the Orbitrap. A full-scan range of MS was set to 400–1500 *m*/*z* at a resolution of 70,000. The scan range of MS/MS started at 100 *m*/*z*, and the resolution was 17,500. A data-dependent acquisition (DDA) mode was used to collect data. Namely, the top 20 peptide parent ions with the highest signal intensity were selected, and fragmented using 30% of the fragmentation energy in the HCD collision pool. Then, MS/MS was performed in the same order. The setting parameters were the following: the automatic gain control (AGC) was 5 × 10^4^, the signal threshold was 63,000 ions/s, the maximum injection time was 80 ms, and the dynamic exclusion time for MS and MS/MS scanning was 30 s.

### 2.8. Data Analysis

We used the Maxquant software (v1.5.2.8) to retrieve MS/MS data. The search parameters were specified as follows: the database was Ovis_aries_Uniprot_9940 (23109 sequences), and a reverse decoy database and a common pollution database were added to calculate the false discovery rate (FDR) caused by random matching and eliminate the effects of contaminated proteins, respectively. Trypsin/P was specified as a cleavage enzyme. Two missed enzymatic cleavage sites were allowed, seven amino acid residues as the minimum peptide, and the maximum modification of the peptide was set to five. Mass tolerance for precursor ions was set as 20 ppm and 5 ppm in the first and main search, respectively. The mass error tolerance for fragment ions was 0.02 Da. Cysteine alkylation was set as a fixed modification, and oxidation of methionine, acetylation and deamination of the N-terminal protein were set as variable modifications. TMT-6 plex was used as a quantitative method, FDR was set as 1%, and unique peptides ≥ 1 per protein for all protein identification.

### 2.9. Bioinformatic Analysis

The UniProt-GOA database (http://www.ebi.ac.uk/GOA/ (accessed on 10 August 2022)) combined with the InterProScan tool (https://www.ebi.ac.uk/interpro/ (accessed on 10 August 2022)) were applied to annotate the Gene Ontology (GO) functional information of proteins, which included three categories: biological process, cellular component and molecular function. Simultaneously, the InterProScan was also used to annotate the proteins’ functional domain based on the protein sequence-alignment method. For Kyoto Encyclopedia of Genes and Genomes (KEGG) pathway enrichment, the KASS service (https://www.genome.jp/tools/kaas/ (accessed on 10 August 2022)), combined with the DAVID tool (https://david.ncifcrf.gov/ (accessed on 10 August 2022)), was performed to annotate the protein information [45]. WoLF PSORT service (https://www.genscript.com/wolf-psort.html (accessed on 14 August 2022)) was used for protein subcellular localization prediction, and STRING (http://www.string-db.org/ (accessed on 14 August 2022)) was conducted to form the protein–protein interaction (PPI) network of the differentially abundant proteins (DAPs). Additionally, hierarchical clustering was performed by the R language. The DAPs were filtered for all categories which were enriched in at least one of the clusters with *p* < 0.05. The filtered *p* value matrix was transformed by the function x = −log10, x values were z-transformed, and z scores were then clustered by one-way hierarchical clustering. The “heatmap.2” from “gplots” R-package was used to visualize cluster membership. Blast tool (http://blast.ncbi.nlm.nih.gov/Blast.cgi (accessed on 18 August 2022)) was performed to give the protein–protein interaction of DAPs at a confidence score > 0.7. The visual diagram was shown by R package “networkD3” (https://cran.r-project.org/web/packages/networkD3/ (accessed on 18 August 2022)).

### 2.10. Parallel Reaction Monitoring (PRM) for Verification

To verify protein expression levels obtained by TMT LC-MS/MS analysis, the expression levels of differentially abundant proteins (unique peptide ≥ 2, FC ≥ 1.5, *p* < 0.05) in two groups were selected randomly and quantified by a PRM assay. Peptides were prepared according to the TMT. Peptide mixtures (1 μg) were introduced into the mass spectrometer via an in-house packed C18 column (Thermo Scientific, 15 cm length, 75 μm i.d.). Chromatographic separation was performed using the EASY-nLC 1000 UPLC system (500 nL/min, Thermo, Waltham, MA, USA). We used one-hour liquid chromatography gradients with acetonitrile ranging from 8 to 35% over 50 min. PRM analysis was performed using a Q-ExactiveTM mass spectrometer (Thermo Fisher Scientific, Bremen, Germany). The mass spectrometer was operated in the positive ion mode with the following parameters: full scan was performed with an *m*/*z* scan range of 350–1000 and the electrospray voltage was 2.0 kV. Intact peptides were detected in the Orbitrap at a resolution of 35,000. Peptides were then selected for MS/MS using the NCE setting of 25, and the fragments were detected in the Orbitrap at a resolution of 17,500. A data-independent procedure that alternated between one MS scan was followed by 20 PRM scans at a resolution of 17,500. Automatic gain control (AGC) was set at 3 × 10^6^ with a maximum injection time of 120 ms. The targeted peptides were isolated within a 1.6 window. Ion activation/dissociation was performed at a normalized collision energy of 25 in higher-energy dissociation (HCD) collision cells. The raw data obtained were analyzed using the Thermo Scientific Proteome Discoverer 1.3 software (Thermo Fisher Scientific, Bremen, Germany). The false discovery rate was set to 0.01 for proteins or peptides, and Skyline software 2.6 (MacCoss Lab, University of Washington) was used for quantitative data processing and proteomic analysis. Peptide settings: enzyme was set as Trypsin [KR/P], and max missed cleavage set as 0. The peptide length was set as 7–25, variable modification was set as Carbamidomethyl on Cys and oxidation on Met, and max variable modifications was set as 3. Transition settings: precursor charges were set as 2, 3, ion charges were set as 1, ion types were set as b, y. The product ions were set from ion 3 to the last ion, and the ion match tolerance was set as 0.02 Da.

### 2.11. Statistical Analysis

IBM SPSS statistics 26 was employed to analyze the data for kinematic parameters of spermatozoa between NC and CAP groups from rams, and a two-tailed Fisher’s exact test with Bonferroni correction (*p* < 0.05) was employed to test the enrichment of DAPs against all identified proteins. Functional enrichment with a corrected *p* < 0.05 was considered significant.

## 3. Results

### 3.1. Evaluation of Capacitation, Tyrosine Phosphorylation Level, and Kinetic Parameters of Sperm before and after Capacitation

Spermatozoa from non-capacitation and capacitation were isolated and pooled for protein extraction, TMT labeling and mass spectrometry analysis. For capacitation, spermatozoa were stained with G-250 staining, using electron microscopes to observe the result of the sperm of G-250 staining, and sperm tails were dyed green to pale blue. The acrosomal region of sperm without capacitation was blue (Figure 1A) and the acrosomal region of sperm after capacitation was either unstained or pale blue (Figure 1B). The sperm acrosome membrane ruptured after AR, and the acrosomal region was dyed in pale blue, due to glycoprotein released. In addition, we counted the number of sperm that has undergone AR (an important indicator of sperm capacitation) per 500 spermatozoa. Three smears were performed per sample. The average AR rate was greater than 60% and sperm capacitation was determined (Appendix A). Also, the expression level of Tyr-P was significantly upregulated (*p* < 0.05) after sperm capacitation (Appendix A, Appendix A). These were evidence of capacitive changes in sperm. Motility kinematic parameters of sperm before and after capacitation were also assessed using the CASAS. Sperm motility (MOT), curvilinear velocity (VCL), straight-linear velocity (VSL) and average path velocity (VAP) were significantly increased (*p* < 0.05) after capacitation, compared to before capacitation. However, the average lateral amplitude of the sperm head (ALH) significantly decreased (*p* < 0.05) after capacitation (Table 1, Appendix A). After capacitation, sperm-tail oscillation amplitude increased, gradually becoming hyperactivated. The acrosome contents of spermatozoa became swollen and the plasma membrane ruptured. Variation in the above parameters also illustrated this point.

The mass spectrometry proteomics information was supplied, regarding data availability: data have been submitted to the ProteomeXchange Consortium (http://proteomecentral.proteomexchange.org/cgi/GetDataset (accessed on 20 August 2022)) via the iProX partner repository, with the dataset identifier PXD023101. A total of 18,894 peptide fragments were identified from 42,404 matched spectrums, with 2378 corresponding proteins in two groups, of which 2050 proteins were quantified (Appendix A and Appendix A). Pearson correlation coefficients of biological replicates showed high correlation for the two group samples (Figure 1C), suggesting the high reproducibility of sample preparation and quality of sequencing data. Principal component (PC) analysis of all detected proteins showed that proteins from two groups were separated into two distinct clusters (Figure 1D), which indicates that non-capacitating and capacitating spermatozoa have significantly different protein abundance.

### 3.2. Identification of Differentially Abundant Proteins

Fold change of the DAPs between the comparable groups (CAP vs. NC) was determined based on the screening criteria group ratio (fold change > 1.2 or <0.83, *p* < 0.05). These DAPs were effectively separated using RStudio (version 3.5.1, 2018), as shown in the volcano plot (Figure 2A); a total of 348 DAPs were identified, where 68 and 280 proteins showed highly expressed abundance in NC and CAP groups, respectively (Appendix A). Among the DAPs, the characterized protein Albumin (ALB) showed the highest relative upregulation and the protein Hemoglobin subunit beta (HBB) showed the highest relative downregulation between the comparable groups. The top 15 upregulated and downregulated proteins of ram spermatozoa in CAP compared to NC are given in Table 2 and Table 3, respectively. These DAPs were mainly distributed in the cytoplasm (37.9%), in extracellular space (21.6%), the nucleus (15.5%), and mitochondria (8.3%) (Figure 2B). Overall categories of these DAPs will help to understand their functions; 284 of the 348 proteins were classified into groups, including post-translational modification (PTM), protein turnover, chaperones, energy production and conversion, intracellular trafficking, secretion, and vesicular transport. Some proteins were classified as having an unknown function (Appendix A, Appendix A).

### 3.3. Validation of Selected DAPs by PRM

For more authentication of TMT LC-MS/MS proteomic data, we performed PRM verification. Raw data of PRM have been submitted to the PRIDE repository (https://www.ebi.ac.uk/pride/ (accessed on 20 August 2022)) and acquired the dataset identifier PXD034330. Here, the peptide information used for PRM quantification is shown in Appendix A, and 11 target DAPs with a change over 1.5-fold and at least two unique peptides for validation, namely, GK2, PSMD4, PSMA6, UCHL5, PSMB5, HBB, SGTA, VCP, SLC25A12, PSMA1 and PSMB3 (Figure 3, Appendix A). Among them, HBB, GK2 and SLC25A12 were downregulated (CAP/NC ratio < 0.67), whereas the expressions of PSMD4, PSMA6, UCHL5, PSMB5, SGTA, and PSMA1 were upregulated (CAP/NC ratio > 1.50). The PRM results showed a similar trend to the TMT results, which indicated that the proteomics data were reliable.

### 3.4. Functional Enrichment Analysis Based on DAPs

To explore the potential roles of DAPs in the comparable group (CAP/NC), we conducted GO terms and KEGG pathway. A total of348 DAPs were mainly clustered into 25 GO categories within ‘biological process’ (BP), ‘cellular component’ (CC), and ‘molecular function’ (MF), based on their GO annotation. Each DAP was assigned more than one term. DAPs were associated with cellular, metabolic, and single-organism processes, biological regulation, response to stimulus, and localization in 13 BP terms; their main enrichment categories included cell, organelle, extracellular region, and membrane in seven CC terms; and these DAPs were mainly involved in binding and catalytic activity in five MF terms (Figure 4A, Appendix A). KEGG was utilized for functional pathway annotation of all DAPs, and only part of the proteins were mapped to the pathways such as four disease-related signaling pathways (Figure 4B, Appendix A). Based on the molecular function in GO enrichment analysis, partial differential proteins were strictly related to ATP production during sperm capacitation, such as ATP6V1H and ATP6V1F (Table 4).

### 3.5. Protein Network Analysis

STRING was used to acquire PPI information of DAPs (*p* < 0.05) and perform the PPI visual network diagram. A total of 81 proteins with the closest interaction relationship with a confidence score > 0.7 (high confidence) were screened out and the protein networks were plotted [46] (Figure 5, Appendix A). Compared to ram spermatozoa before capacitation, most of the functional modules were upregulated after capacitation, except for NDUFAB1, NDUFS4, UQCRQ, ATP5IF1, ATP5PD and W5NZM0 (uncharacterized protein in database). Furthermore, significantly positive associations between PSMA2, PSMB7, UBA52, NEDD8, RAD23B, PPIB, NT5E, UGP2, SORD, and W5Q420 (uncharacterized protein in database) abundance and spermatozoa capacitation were observed (Log2Ratio > 1.01). For the PPI degree of connectivity, 13 target proteins with a degree of connectivity over 25 were collected, such as PSMD2, PSMD4, PSMD1, PSMA5, PSMA2, and PSMA1, which had a strong interaction among themselves and were annotated to the ubiquitination functional pathway (All Log2Ratio > 0.59).

## 4. Discussion

As a highly specialized cell, sperm is thought to be largely quiescent in terms of transcriptional and translational activity. As a result, once it has left the male reproductive tract, the sperm is essentially operating with a static population of proteins, with functions such as post-translational modification (PTM) of proteins as one of its specific functions before fertilizing the oocyte [47,48]. Typically, capacitation begins with the removal of sperm plasma-membrane stabilizing factors that trigger an efflux of cholesterol, calcium and bicarbonate influx, and an increase in intracellular pH. These events result in an elevation in the protein tyrosine phosphorylation and sperm hyperactivation [33,49,50]. Based on our result, the level of the protein tyrosine phosphorylation was increased after capacitation, which provided evidence for these capacitive changes in ram spermatozoa.

However, given the limited understanding of the capacitation process, deciphering the molecular mechanism involved remains the research focus in the field of reproduction. Known proteomic techniques such as MS, iTRAQ and TMT, are credible strategies to help investigate the alteration of sperm components during sperm capacitation, and the molecular mechanisms have been elucidated in humans [10,17], mice [12,19], and boar [51]. Zhang et al. investigated proteomic profiling in buffalo spermatozoa before and after capacitation using the TMT strategy: 1461 proteins were quantified, and 93 DAPs were screened. These DAPs were mainly involved in pathways of oxidative phosphorylation, metabolic and PPAR signaling pathways, and acquired marker proteins such as CYLC1, CYLC2, SLC26A8, and VAMP4 [41]. In addition, Peris-Frau et al. compared the changes in protein profile in fresh vs. cryopreservation Manchega ram sperm during in vitro capacitation by RP-LC-MS/MS; results indicated that the expression abundances of 14 proteins related to mitochondrial activity, sperm motility, apoptosis-stress response, etc., were significantly different [26,52]. These findings provide insights into acquiring marker proteins and the mechanisms that are associated with sperm capacitation at the molecular level. In this study, we monitored the dynamic changes in protein expression levels during ram sperm capacitation in vitro, using the LC-MS/MS. A total of 2050 proteins were quantified, including 348 DAPs that are 280 upregulated and 68 that are downregulated after capacitation, respectively. In the clusters of orthologous group (COG) categories of those DAPs, 284 were classified into groups, including PTM, protein turnover, chaperones, energy production and conversion, inorganic ion transport and metabolism, intracellular trafficking, secretion, vesicular transport, cytoskeleton, etc. This was consistent with the published reports that mammalian sperm capacitation-related proteins are categorized into multiple groups, namely, sperm motility and differentiation-related proteins (including energy-related enzymes, structural proteins, activator signal transducers), sperm–zona pellucida interaction and sperm–oolemma penetration proteins, acrosome biogenesis and acrosome-reaction proteins, nuclear proteins, peripheral proteins, and PTM proteins [53]. Interestingly, proteins like PGP (phosphoglycolate phosphatase), which was upregulated during ram spermatozoa capacitation in vitro, categorized into inorganic ion transport and metabolism. It catalyzes glycerol production from glycerol-3-phosphate, a metabolic intermediate of glucose, lipid and energy metabolism. It also has protein tyrosine phosphatase activity, for protein tyrosine phosphorylation, a cAMP-dependent pathway that operates in capacitated spermatozoa [54]. Partial proteins strictly related to sperm capacitation with a significant difference between the two groups (CAP vs. NC), such as ALB, with the highest upregulated abundance, have been shown to extract cholesterol out of the sperm cell membrane with an HCO3-dependent mode, which can activate protein kinase A (PKA) during capacitation; this results in a reorganization of the phospholipid distribution in the cell membrane bilayer, thereby facilitating cholesterol depletion by ALB (Hereng et al., 2014). HBB was a DAP with the highest downregulated abundance, and could facilitate intracellular transport and storage of oxygen (O_2_) in sperm. Downregulation of HBB might reduce the transport of O_2_ to the outside of cells, which facilitates ATP production [55]. This could directly affect the fertilization potential, due to the increase in sperm motility, and could affect the calcium pathway, activating capacitation and the subsequent acrosome reaction [56].

We found that several PSMs (the 26S proteasome), such as PSMA2 and PSMB7, were significantly upregulated in the ram capacitation sperm; PSMD2, PSMD4, PSMD1, PSMA5, and PSMA1 had a PPI degree of connectivity greater than 25. Our results corresponded with a previous study conducted by Choi YJ et al., in which PSMA4, PSMB1 and PSMB10 were upregulated in boar sperm capacitation [57]. Ubiquitination is one of the canonical PTMs of target proteins, using covalent attachment of the small chaperone protein ubiquitin. In general, the ubiquitin proteasome system (UPS) is responsible for the degradation and turnover of abnormal, short-lived or damaged proteins in eukaryotic cells. It involves the remodeling of sperm plasma membrane and acrosome during sperm capacitation, and sperm–ZP interaction [57]. Conventional UPS covalently modifies substrate proteins with poly-ubiquitin chains and targets them for recognition and degradation by PSM; a holoenzyme with specific affinity for ubiquitinated proteins, it is involved in many regulatory pathways, comprising a 20S core and a peripheral 19S regulatory particle. During capacitation, the 26S proteasome could be activated by PKA, and the substrate protein is recognized by the 19S regulatory particle and phosphorylation at serine/threonine residues through a feed-back loop [58]. In boar spermatozoa, subunits PSMA1-7 (Proteasome 20S Subunit Alpha 1-7) of 20S were not post-translationally modified, with the exception of subunit PSMD4 (Proteasome 26S Subunit, Non-ATPase 4) of 19S during the course of capacitation [59]. Other 26S proteasome non-ATPase regulatory subunits, such as PSMD13 (Proteasome 26S Subunit, Non-ATPase 13) could negatively mediate the degradation of mutant membrane-associated proteins by the ubiquitination-mediated proteasomal pathway [60]. PSMD6 (Proteasome 26S Subunit, Non-ATPase 6) maintains the integrity of the 26S complex by maintaining the lid structure [61]. Alternatively, PSMB7 (Proteasome 20S Subunit Beta 7) is also important for proteasome assembly. PSMB5 precursors exist in a series of stage-specific spermatogenic cells [62], and are modulated by small GTPases, which facilitate cytoskeleton remodeling in the AR process [63]. We found that the abundance of DAPs such as PSMA1, PSMA5, PSMB2, PSMB4, PSMB5, PSMB7, PSMD1, PSMD2, PSMD4, PSMD6, and PSMD13 were higher in ram spermatozoa after capacitation than before capacitation. These PSM families were classified into the category of PTM, protein turnover, and chaperones, although the description of PSMD6 and PSMD13 were uncharacterized in Ovis aries. These results indicate the benefits of these subunits in accelerating sperm capacitation through ubiquitin proteasome pathways.

Other proteins involved in energy production and conversion were mainly located in the nucleus (PRUNE1), cytoplasm (GALT, ACLY, ATP6V1H, ATP6V1F, ATP5PD, ATP5IF1) and mitochondria (ATP6V1B2). Except for ATP5PD and ATP5IF1, they were more abundant in ram spermatozoa following capacitation compared with before. ATP5PD (ATP Synthase Peripheral Stalk Subunit D), a mitochondrial membrane ATP synthase (F1F0-ATP synthase), catalyzes ATP synthesis from ADP in the presence of an electrochemical gradient of protons across the inner membrane during oxidative phosphorylation. ATP5PD leads to ROS accumulation under normoxia [64]. Downregulation of ATP5PD helps maintain a small amount of ROS during sperm capacitation. Additionally, ATP5IF1 (ATP Synthase Inhibitory Factor Subunit 1) is the most characterized regulator of F1F0-ATP synthase [65], and binds to the enzyme which limits ATP hydrolysis when the mitochondrial membrane potential falls below a threshold. Downregulated expression of ATP5IF1 and F1F0-ATP synthase during energized conditions, and other specific agonists (such as A2AdoR), stimulate the activation of sperm soluble adenylyl cyclase, which converts ATP to cAMP in non-capacitating sperm; an increase in cAMP activates PKA activity, and accelerates sperm capacitation in mammals [66,67]. PRUNE1 (Prune Exopolyphosphatase 1) belongs to the phosphoesterase DHH superfamily-pertaining protein, and has cyclic nucleotide phosphodiesterase (PDE) activity [68]. PDEs are known to modulate the biological availability of cAMP by hydrolyzing the 3′-5′ phosphodiester bond [69]. PRUNE1 enriched in ram spermatozoa after capacitation was discovered, indicating its likely role in the regulation of cAMP during sperm capacitation. It is generally known that vacuolar-type H+ ATPase (V-ATPase) is a highly conserved enzyme among all the eukaryotic cells, where it couples the energy of ATP hydrolysis to proton transport across vesicular and plasma membranes, and mediates acidification in organelles [70]: for example, the accumulation of plasma membrane V-ATPase in narrow and clear cells of epididymis and vas deferens, which participates in active proton transport and pH adjustment of intracellular compartments [71]. Subunits of V-ATPase existed in ram spermatozoa, including ATP6V1H (ATPase H+ Transporting V1 Subunit H), ATP6V1F (ATPase H+ Transporting V1 Subunit F) and ATP6V1B2 (ATPase H+ Transporting V1 Subunit B2), all of which were upregulated following capacitation. They belong to the V1 domain, and are mainly involved in ATP hydrolysis [72]. ATP and its hydrolysis product adenosine are essential components for the formation of G-protein-coupled proteins (AdoR). Activating the stimulatory region therein (A2a, A2b) increases the second-messenger levels, thereby regulating cAMP levels and activating PKA activity in the spermatozoa [66]. Likewise, GALT (Galactose-1-Phosphate Uridylyltransferase) and ACLY (ATP Citrate Lyase) are related to energy production and conversion, and they are upregulated in sperm after capacitation. GALT catalyzes the second step of the Leloir pathway of galactose metabolism, and is an important component for sperm binding to ZP [73]. Thus, the abundance of GALT after capacitation initiates subsequent AR and sperm–zona binding [74]. ACLY is also an enzyme responsible for the formation of acetyl-CoA and oxaloacetate from CoA and citrate, with the concomitant hydrolysis of ATP to ADP and phosphate. Acetyl-CoA serves fatty acid and cholesterol synthesis. Fatty acid oxidation activity increases when sperm capacitation starts, and can produce ATP, providing additional metabolic fuel for triggering the capacitation process [75].

Transport and metabolism during sperm capacitation are also of great significance. Proteins such as GNPDA1 (Glucosamine-6-Phosphate Deaminase 1) and GNPDA2 (Glucosamine-6-Phosphate Deaminase 2), which are involved in carbohydrate transport and metabolism, were enriched in ram spermatozoa after capacitation. GNPDA (GNPDA1 and 2) convert GlcN6P back to Fru6P to synthesize UDP-GlcNAc, a nucleotide sugar that serves as a substrate for the synthesis of glycoconjugates like hyaluronan [76]. We have also identified the fact that PPP4R1 was involved in signal transduction mechanisms. PPP4R1 usually forms the PP4c/PP4R1 complex that is implicated in the regulation of histone acetylation and DNA-damage checkpoint signaling [77]. This uniquely copurifies with HDAC3, and displays protein phosphatase activity [78].

## 5. Conclusions

In this paper, changes in protein components in ram spermatozoa under non-capacitating and capacitating conditions in vitro were investigated by a TMT-based proteomics approach. The functional enrichment showed that these identified differentially abundant proteins were annotated to some essential pathways such as ubiquitination (PSM family like PSMA1, PSMA5, PSMB2, PSMB4, and PSMB5), energy production and conversion (PRUNE1, GALT, ACLY, ATP6V1H, ATP6V1F, ATP5PD, and ATP5IF1), and diverse transport and metabolism (PGP, GNPDA1, GNPDA2 and PPP4R1) during sperm capacitation. Whilst these protein alterations offer potential novel biomarkers, the specific molecular mechanism regulating sperm capacitation deserves further study.

## Figures and Tables

**Figure 1 animals-14-02363-f001:**
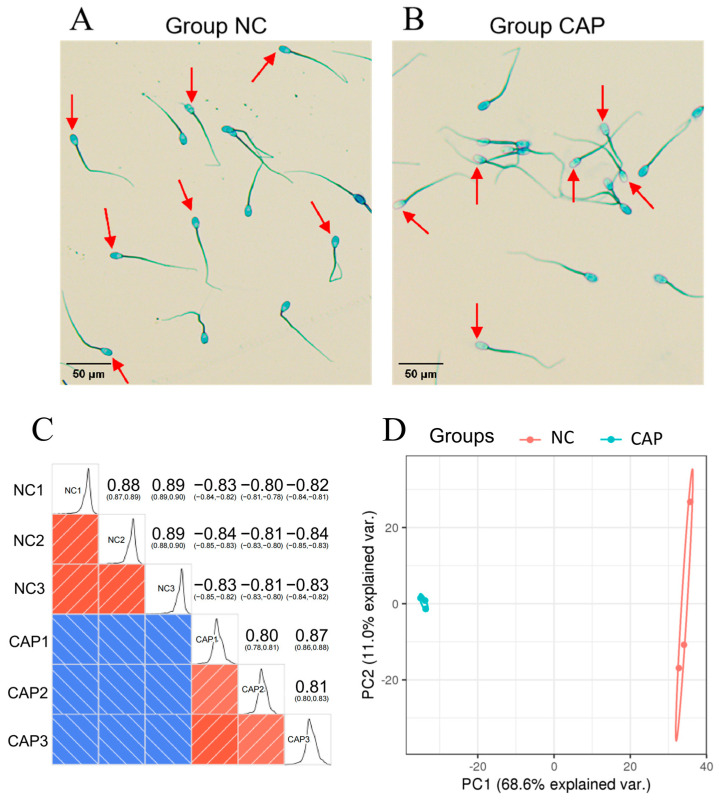
Proteomics analysis of ram spermatozoa before and after capacitation in vitro. Coomassie brilliant blue staining observed on (**A**) non-capacitating spermatozoa with blue over entire head and (**B**) capacitating spermatozoa with no staining or a light blue in acrosome region (bar = 50 µm) before sequencing; acrosome regions are indicated by red arrows. (**C**) Pearson correlation coefficient quantitative results of biological or technical duplicate samples of non-capacitating (NC) and capacitating (CAP) groups. (**D**) Principal component (PC) analysis of the proteomics for the two groups are shown by symbols of the same color. PC1 and PC2 represent the top two dimensions of detected proteins among spermatozoa.

**Figure 2 animals-14-02363-f002:**
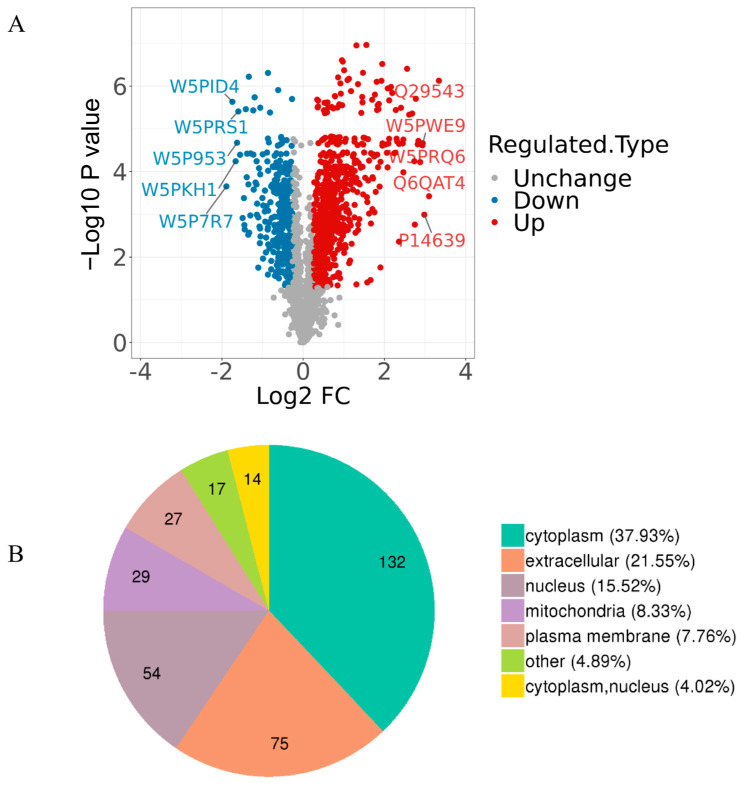
Screening of the DAPs. (**A**) Volcano plots of the comparison (CAP vs. NC). The horizontal coordinate represents the fold change of the DAPs (log2). The vertical coordinate represents the *p*-value (10 is the logarithmic transformation at the bottom). Red points indicate significantly upregulated DAPs; blue points indicate significantly downregulated DAPs, and gray points indicate proteins that did not show differential abundance. (**B**) Subcellular localization of 348 DAPs that were identified in the comparable group (CAP vs. NC).

**Figure 3 animals-14-02363-f003:**
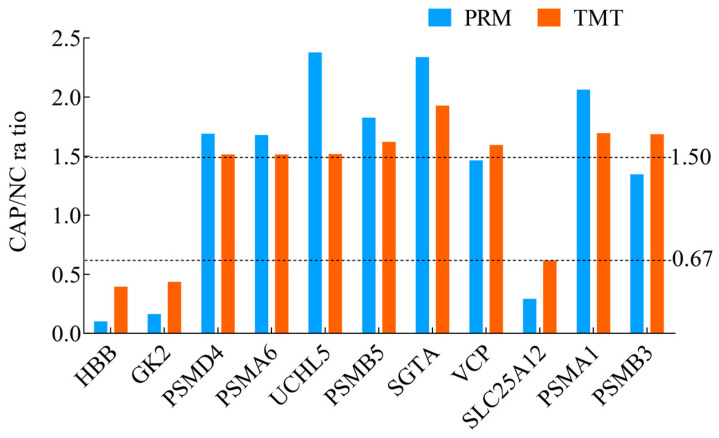
Expression patterns of selected DAPs using TMT analysis and PRM validation. From top to bottom, two dotted lines represent 1.5-fold (upregulation) and 0.67-fold (downregulation), respectively.

**Figure 4 animals-14-02363-f004:**
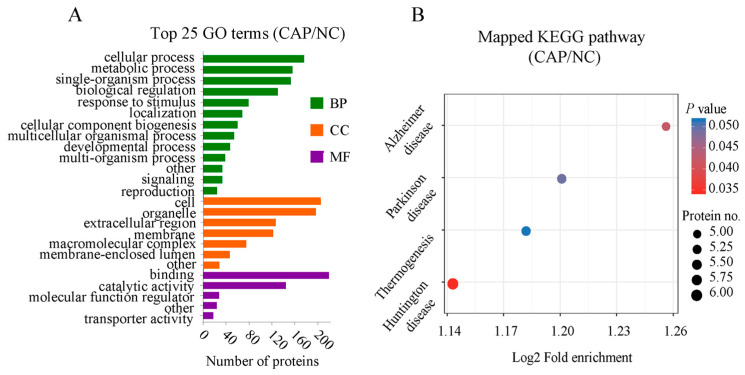
Functional enrichment analysis of DAPs. (**A**) GO analysis for DAPs. Three functional domains are displayed, including biological process, cellular component and molecular function with the top 13 terms, 7 terms and 5 terms, respectively. (**B**) Analysis of KEGG pathways in the comparable groups (CAP vs. NC).

**Figure 5 animals-14-02363-f005:**
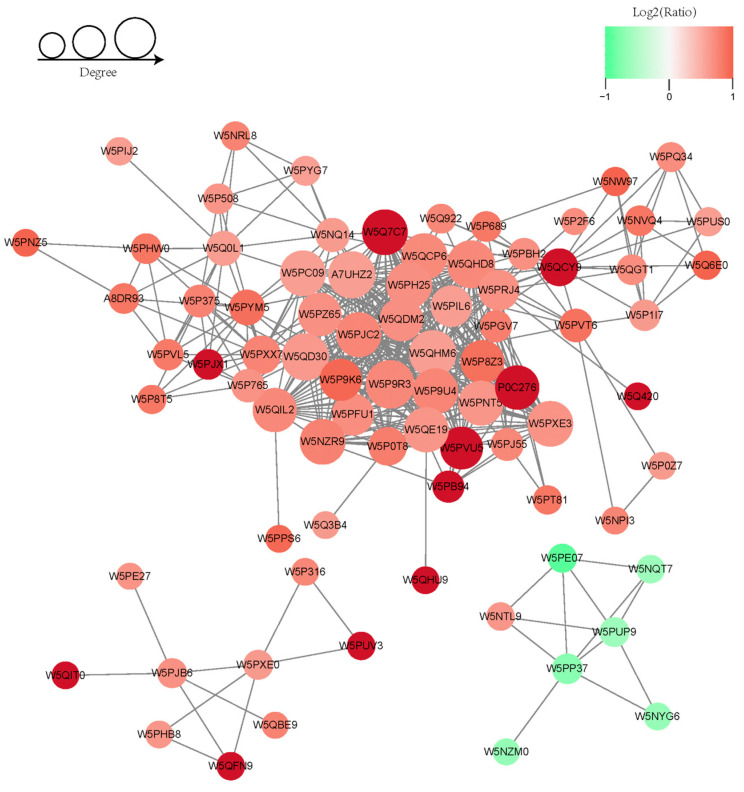
Protein network analysis of 81 DAPs with the closest interaction relationship, with a confidence score > 0.7 (high confidence) in two groups.

**Table 1 animals-14-02363-t001:** Kinematic parameters of spermatozoa between NC and CAP groups from rams.

Sperm Kinematics	Group NC	Group CAP
MOT (%)	79.3 ± 6.4	83.5 ± 4.3 *
VCL (μm/s)	113.2 ± 11.2	123.5 ± 4.0 *
VSL (μm/s)	38.4 ± 8.1	68.2 ± 4.9 *
VAP (μm/s)	66.6 ± 10.2	96.7 ± 4.4 *
ALH (μm)	4.3 ± 0.5	3.2 ± 0.2 *

Note: * indicates significant difference (*p* < 0.05) of sperm between NC and CAP groups.

**Table 2 animals-14-02363-t002:** The top 15 proteins that were more abundant in NC compared to CAP.

Accession	Gene Name	MW [kDa]	Score	Coverage [%]	Peptides	*p*-Value	Fold Change
P02075	HBB	16.07	142.24	74.5	11	6.03 × 10^−7^	2.53
W5Q3H2	DNAL4	12.97	27.66	33.9	4	3.46 × 10^−6^	2.66
W5NSD8	TEPP	30.63	87.98	29.6	10	3.72 × 10^−5^	1.57
W5NZ60	GNPAT	77.68	32.77	8.2	5	0.000324	1.57
W5P9X2	CHCHD3	26.76	125.87	40.9	9	0.000579	1.52
W5PTU5	SPACA1	32.33	323.31	46.9	11	0.000644	1.87
W5NQT7	NDUFAB1	17.59	33.54	21	4	0.001137	1.53
B2MVX2	SLC25A11	34.21	27.07	12.1	3	0.002582	1.634
W5P657	WDR87	331.92	35.97	2	6	0.003284	2.07
W5P066	CYB5B	16.98	50.35	13.7	2	0.004602	1.61
W5Q3Y4	ATP2A2	106.61	35.55	5.8	6	0.004741	2.09
W5Q4Y3	STPG2	61.29	25.99	5.3	2	0.005075	1.57
W5PP37	ATP5PD	18.72	45.97	39.1	7	0.005245	1.63
W5QHI0	IMMT	89.27	323.31	35.5	28	0.008158	1.52
W5PW99	ADCK5	63.79	20.70	5.5	3	0.017699	2.15

**Table 3 animals-14-02363-t003:** The top 15 proteins that were more abundant in CAP compared to NC.

Accession	Gene Name	MW [kDa]	Score	Coverage [%]	Peptides	*p*-Value	Fold Change
P14639	ALB	69.19	323.31	53.7	31	0.001021	7.89
W5P7X5	WFDC2	10.47	171.69	76	11	2.3 × 10^−5^	7.05
W5QB42	CES5A	64.21	119.1	21	11	0.004365	5.13
W5P1C2	NUCB2	52.47	65.79	22.4	8	2.15 × 10^−6^	4.34
W5Q0Z2	CTSB	36.73	51.60	17.3	6	2.21 × 10^−5^	4.27
W5Q9A2	AZGP1	35.42	61.43	23.3	5	2.47 × 10^−5^	3.86
W5PR14	FAM3D	25.39	20.86	14.3	3	0.000224	3.68
W5Q1M0	GLB1	72.84	82.49	15.1	8	3.46 × 10^−6^	3.61
W5NQ55	DEFB116	11.71	69.49	30.1	3	4.02 × 10^−5^	3.54
P56283	NPPC	13.32	38.63	34.1	4	4.42 × 10^−5^	3.36
W5QHU9	PPIB	23.73	50.51	32.4	6	3.95 × 10^−5^	3.31
H9TN93	GPx5	25.04	133.3	54.3	13	1.74 × 10^−5^	3.02
W5PD71	CRP	25.27	251.4	33.5	7	3.57 × 10^−5^	2.83
W5QIT0	SORD	38.27	168.92	40.2	15	0.001724	2.56
W5PTV7	ACE	138.23	229.67	17	21	8.97 × 10^−7^	2.55

**Table 4 animals-14-02363-t004:** Major differential proteins related to ram sperm capacitation in vitro.

Accession	Gene Name	Peptides	MW [kDa]	Score	Coverage [%]	CAP/NC	*p*-Value
W5QHM6	PSMD1	22	105.95	234.400	28.5	1.511	7.48 × 10^−3^
A7UHZ2	PSMD4	8	40.78	87.409	30.5	1.515	1.42 × 10^−3^
W5QHD8	PSMD2	18	107.61	158.900	22.1	1.663	2.62 × 10^−4^
W5QD30	PSMA5	8	26.41	124.850	46.5	1.545	1.98 × 10^−4^
W5QE19	PSMB2	6	22.97	43.467	27.2	1.575	2.57 × 10^−4^
W5QIL2	PSMB4	4	28.97	54.891	25.0	1.667	2.39 × 10^−4^
W5QDM2	PSMB5	6	28.71	47.961	24.3	1.620	4.30 × 10^−3^
W5PFU1	PSMD13	15	42.75	122.600	38.3	1.624	2.01 × 10^−5^
W5PVU5	PSMB7	3	29.92	25.420	8.3	2.030	5.68 × 10^−3^
W5P9U4	PSMA1	12	29.59	101.290	41.1	1.695	7.68 × 10^−5^
W5PNT5	PSMD6	13	45.51	130.610	32.9	1.570	3.19 × 10^−4^
W5QIC3	PRUNE1	7	48.85	57.096	20.1	1.580	1.00 × 10^−3^
W5PHB8	GALT	7	43.34	48.862	16.9	1.567	3.94 × 10^−3^
Q2TCH3	ACLY	29	120.93	236.510	30.3	1.944	2.80 × 10^−6^
W5PWF2	ATP6V1H	10	55.80	123.990	29.4	1.614	1.50 × 10^−5^
W5NZZ3	ATP6V1F	6	13.40	36.524	42.0	1.622	3.16 × 10^−3^
W5PP37	ATP5PD	7	18.72	45.973	39.1	0.612	5.25 × 10^−3^
W5NYG6	ATP5IF1	4	11.37	44.214	19.8	0.631	2.41 × 10^−2^
W5PLZ0	ATP6V1B2	19	56.93	193.890	43.7	1.792	2.11 × 10^−5^
W5Q8X9	PGP	9	25.01	129.770	41.8	1.542	1.98 × 10^−4^
W5NUG3	GNPDA1	8	32.56	40.061	33.2	1.513	5.18 × 10^−4^
W5Q0A8	GNPDA2	10	31.16	109.880	43.5	1.770	1.34 × 10^−3^
W5QFN9	UGP2	15	56.90	158.290	33.9	2.015	8.29 × 10^−5^
W5NSS1	PPP4R1	12	105.74	111.110	17.5	1.805	1.86 × 10^−5^
W5NZ60	GNPAT	5	77.68	32.765	8.2	0.637	3.24 × 10^−4^

## Data Availability

The data generated by this project are included as Appendix A. Requests for raw data of TMT labeling strategy could search via the iProX partner repository with the dataset identifier PXD023101, and raw data of PRM have also been submitted to the PRIDE repository (https://www.ebi.ac.uk/pride/ (accessed on 20 August 2022)) with the dataset identifier PXD034330. All requests for data could be made to the corresponding authors.

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
