# Peer review of "Comparative Proteomic Identification of Ram Sperm before and after In Vitro Capacitation"

_animals, 2024, doi:10.3390/ani14162363_

Round 1

Reviewer 1 Report

Comments and Suggestions for Authors

A very interesting and relevant article, but there are questions about the design of the experiment. The file with additional materiales is not available, so it is not clear what "pellets" (lines 118, 130) and "frozen pellets" (line 142) mean? Was the sperm frozen? I think the Materials and Methods section needs to be improved.

Author Response

Point 1: A very interesting and relevant article, but there are questions about the design of the experiment. The file with additional materiales is not available, so it is not clear what "pellets" (lines 118, 130) and "frozen pellets" (line 142) mean? Was the sperm frozen? I think the Materials and Methods section needs to be improved.

Response 1: Thank you for your valuable suggestion on our manuscript. It’s very helpful for improving our manuscript. We added the missing supplementary material. The “pellets” in the article is ambiguous and we have re-written this part according to the Reviewer’s suggestion. In fact, the collected fresh ejaculates were centrifuged to obtain sperm and seminal plasma, and the separated sperm were cleaned with PBS for several times and divided into two parts: NC group and CAP group. The CAP group was added with capacitation solution, and the two groups were incubated together for 30 min to induce an AR, and both groups were centrifuged After incubation. The precipitated part (capacitated sperm) was stored at -80℃ for further experiments. Please also see details in materials and methods 2.2 (Line 114) and 2.3(Line 132).

Reviewer 2 Report

Comments and Suggestions for Authors

In the present work, the authors perform a comparative proteomic work on ram sperm proteins before and after capacitation. The study used an LCMS/MS strategy followed by TMT. The analysis indicates a functional enrichment of the differential abundance of proteins related to energy production, metabolism, RNA processing and vesicular transport pathways. Of these, 280 proteins are upregulated and 68 downregulated. The work is interesting, but there are some points that the authors should consider before publishing the manuscript:

1. The term Droper is rare and does not appear in PubMed. It is crucial to clarify that Droper refers to a ram from the South African region, and therefore, the generic term 'ram' should be used in the text to ensure accurate and understandable communication.

2. Line 264 refers to optical microscopes or electron microscopes.

3. Supplementary figures and tables are not shown. The link leads to the following notice:

Error 404 - File not found

The web page you are looking for could not be found. The URL may have been incorrectly typed, or the page may have been moved into another part of the mdpi.com site.

Include the supplementary figures and tables, as they significantly enhance the analysis and understanding of the manuscript, thereby improving its overall quality.

4. Figure 5A and B: The characteristics described for sperm are challenging to observe since they are minimal, and one cannot differentiate between an non-capacitated sperm and a capacitated one. It would help to show the figure at a higher magnification and with more contrast.

5. Lanes 383-384: As a highly specialized cell, sperm is thought to be largely quiescent in terms of transcriptional and translational activity.

If this is so, what is the mechanism for the upregulation of 280 proteins? To define whether this is due to de novo protein synthesis, assays to inhibit protein synthesis should be carried out or discuss a mechanism that explains the protein upregulation.

6. Figure 5 presents a protein network analysis that could be highly relevant to the capacitation process. It is crucial to discuss the importance of this analysis in the context of sperm physiology, as it provides valuable insights into the capacitation process.

Author Response

Thank you very much for your very professional comments on our manuscript. Those comments are valuable and very helpful for revising and improving our manuscript. We have studied comments carefully and made corrections in this revision. Revised portions are marked with red font in the manuscript. We provide a Comments-by-Comments response to the reviewer’s comments, please see the details below.

Comments 1: The term Droper is rare and does not appear in PubMed. It is crucial to clarify that Droper refers to a ram from the South African region, and therefore, the generic term 'ram' should be used in the text to ensure accurate and understandable communication.

Response 1: We agreed with your Comments and used ram to replace all Droper ram in the text.

Comments 2: Line 264 refers to optical microscopes or electron microscopes.

Response 2: We using electron microscopes to observe the result of the sperm of G-250 staining. Please also see details in Lines 285-288 in this revision.

Comments 3: Supplementary figures and tables are not shown. The link leads to the following notice:

Error 404 - File not found

The web page you are looking for could not be found. The URL may have been incorrectly typed, or the page may have been moved into another part of the mdpi.com site.

Response 3: We added the missing supplementary material.

Comments 4: Figure 5A and B: The characteristics described for sperm are challenging to observe since they are minimal, and one cannot differentiate between an non-capacitated sperm and a capacitated one. It would help to show the figure at a higher magnification and with more contrast.

Response 4: We did not find Figure 5A and B in this article, so we infer that what the reviewer said should be Figure 1A and B, and we improved the resolution of Figure and also enlarged them in the manuscript. Please also see details in Lines 317 in this revision.

Comments 5: Lanes 383-384: As a highly specialized cell, sperm is thought to be largely quiescent in terms of transcriptional and translational activity. If this is so, what is the mechanism for the upregulation of 280 proteins? To define whether this is due to de novo protein synthesis, assays to inhibit protein synthesis should be carried out or discuss a mechanism that explains the protein upregulation.

Response 5: Thank you so much for your comments and good suggestions, which are very helpful for improving our manuscript. As we all known that mammalian sperm is a highly specialized cell, which thought to be largely quiescent in terms of transcriptional and translational activity after leaving the male reproductive tract. On their way through the female reproductive tract (FRT), sperm must bind reversibly with numerous epithelial surfaces simultaneously circumventing the phagocytes, antibodies and complement proteins [1]. The capacitation in the FRT drastically remodel the testicular sperm surface which endows these spermatozoa with motility and competence for fertilization [2]. According to previous studies, the upregulation of protein expression after sperm capacitation may be related to two aspects. Some scholars have suggested that translation of new proteins from mRNA transcripts can take place during sperm capacitation [3]. Zhu Z et al. found that gene expression and protein synthesis in mitochondria improved the duration of high-speed linear motility in boar sperm [4]. This may indicate that the translation of mRNA transcripts changes sperm protein expression. Meanwhile, it has also been proposed that sperm capacitation is mainly dependent on post-translational modifications, such as phosphorylation of preexisting sperm proteins, and phosphorylation of tyrosine residues is a consistent indicator of sperm capacitation [5]. Many studies have shown that capacitation may lead to increased tyrosine phosphorylation, and conversely inhibition of tyrosine phosphorylation can block sperm capacitation, acrosome reactions, and in vitro fertili-zation [6-8]. We have performed relevant discussion and description in the Discussion section of “Revised manuscript”, Please also see details in Lines 445-451 in this revision.

References:

[1]. Archana, S. S., Selvaraju, S., Binsila, B. K., Arangasamy, A., and Krawetz, S. A. (2019). Immune regulatory molecules as modifiers of semen and fertility: a review. Mol. reproduction Dev. 86 (11), 1485–1504.

[2]. Bianchi, E., and Wright, G. J. (2016). Sperm meets egg: the genetics of mammalian fertilization. Annual Rev. Genet. 50, 93–111.

[3]. Cohen, R.; Buttke, D.E.; Asano, A.; Mukai, C.; Nelson, J.L.; Ren, D.J.; Miller, R.J.; Cohen-Kutner, M.; Atlas, D.; Travis, A.J. Lipid Modulation of Calcium Flux through Ca(V)2.3 Regulates Acrosome Exocytosis and Fertilization. Dev. Cell. 2014, 28(3), 310-321.

[4]. Zhu Z, Umehara T, Okazaki T, et al. Gene Expression and Protein Synthesis in Mitochondria Enhance the Duration of High-Speed Linear Motility in Boar Sperm. Front Physiol. 2019;10:252. Published 2019 Mar 12.

[5]. Arcelay, E.; Salicioni, A.M.; Wertheimer, E.; Visconti, P.E. Identification of proteins undergoing tyrosine phosphorylation during mouse sperm capacitation. Int. J. Dev. Biol. 2008, 52(5-6), 463-472.

[6]. Peris-Frau, P.; Martin-Maestro, A.; Iniesta-Cuerda, M.; Sanchez-Ajofrin, I.; Mateos-Hernandez, L.; Garde, J.J.; Villar, M.; Soler, A.J. Freezing-Thawing Procedures Remodel the Proteome of Ram Sperm before and after In Vitro Capacitation. Int. J. Mol. Sci. 2019, 20(18), 4596.

[7]. Wright, P.C.; Noirel, J.; Ow, S.Y.; Fazeli, A. A review of current proteomics technologies with a survey on their widespread use in reproductive biology investigations. Theriogenology. 2012, 77(4), 738-765.e52.

[8]. Alvau A, Battistone MA, Gervasi MG, et al. The tyrosine kinase FER is responsible for the capacitation-associated increase in tyrosine phosphorylation in murine sperm. Development. 2016;143(13):2325-2333.

Comments 6: Figure 5 presents a protein network analysis that could be highly relevant to the capacitation process. It is crucial to discuss the importance of this analysis in the context of sperm physiology, as it provides valuable insights into the capacitation process.

Response 6: Thank you very much for your good comments. We have performed relevant discussion and description in the Discussion section of “Revised manuscript”, Please also see details in Lines 463-468 in this revision.

Reviewer 3 Report

Comments and Suggestions for Authors

The present study aimed to investigate the difference in protein profile of ram sperm before and after capacitation, with a purpose of disclosing the mechanism of sperm capacitation. The results of comparative proteomic analysis indicated that differentially expressed sperm proteins before and after capacitation were involved in various biological process. The most significant proteins found in this study could be potential targets that could be used to study the mechanism of sperm capacitation. This study provides potential proteins candidates that could be involved in sperm capacitation, which are of great importance for the further research on that. There are several points regarding the contents of this study I would like to share my opinions with the authors.

General comments:

This study showed a good presentation of the results. However, when talking about the proteomic differences, it is suggested to state clearly how the difference happened. As the sperm chromosome is highly compacted and almost silent in transcription and translation processes, why certain proteins expressed more or expressed less due to capacitation? In the manuscript, the authors indicated that capacitation induced protein modification. That is right. But how to explain the higher expression of those proteins?

Specific comments:

Q1. In the Introduction section, I suggest to talk more about the progress on the mechanism of sperm capacitation and the problems that still need to solve, and then talk about the results or potential speculations based on proteomic analysis. Finally, the authors are suggested to highlight the innovative point in this study.

Q2. Line 99, two ejaculates from each ram were collected and mixed to make a pooled sample. How did the authors collect the two ejaculates? The time interval between the two ejaculates? How did the authors handle with the semen after collection? Were they stored? How?

Q3. Line 125, it is necessary to state clearly the incubation time and temperature for AR induction.

Q4. Line 114, 127, I suppose the authors would like to express “sperm were wash with PBS by centrifugation (xxx g, xx min)”. What is the reason that the authors used different centrifugation forces and time?

Q5. Line 139-140, it is necessary to add the dilution rate of those antibodies used.

Q6. Line 142, instead of 30 mg frozen sperm pellet, I would prefer to indicate the total sperm number for each sample because that will be better controlled and easy to be duplicated. Furthermore, when the comparison of proteins profiles between samples before and after capacitation, it is reasonable to compare that among the samples of same sperm number.

Q7. Line 291-292 were from the template and should be deleted.

Comments on the Quality of English Language

minor

Author Response

Comments 1: In the Introduction section, I suggest to talk more about the progress on the mechanism of sperm capacitation and the problems that still need to solve, and then talk about the results or potential speculations based on proteomic analysis. Finally, the authors are suggested to highlight the innovative Comments in this study.

Response 1: We have re-written this part according to the Reviewer’s suggestion. Revised portions are marked with red font in the manuscript. Please also see details in Lines 51-101 in this revision.

Comments 2: Line 99, two ejaculates from each ram were collected and mixed to make a pooled sample. How did the authors collect the two ejaculates? The time interval between the two ejaculates? How did the authors handle with the semen after collection? Were they stored? How?

Response 2:  Thank you so much for your comments and good suggestions, which are very helpful for improving our manuscript. Semen samples were collected by means of an artificial vagina (AV) [1]. According to previous studies, the highest fertility rates were obtained when males had generally in two abstinence days [2], so we performed a second sperm collection on the same rams two days after the first ejaculation. After collected fresh ejaculate, we immediately measured sperm motility and density with CASA. Three out of the nine samples were mixed randomly in equal volumes to form a biosample, a total of three mixed biosamples were used in our experimental replication (n = 3). Then, isolated sperm and seminal plasma, and the obtained sperm were divided into two parts: one part was used as non-capacitating (NC) sample resource, the other part was used for capacitation. After the capacitation test, it was stored at -80℃.

References:

[1]. Maroto-Morales A, Ramón M, García-Alvarez O, et al. Characterization of ram (Ovis aries) sperm head morphometry using the Sperm-Class Analyzer. Theriogenology. 2010;73(4):437-448.

[2]. Montes-Garrido R, Riesco MF, Anel-Lopez L, et al. Application of ultrasound technique to evaluate the testicular function and its correlation to the sperm quality after different collection frequency in rams. Front Vet Sci. 2022;9:1035036. Published 2022 Nov 25.

Comments 3: Line 125, it is necessary to state clearly the incubation time and temperature for AR induction.

Response 3: Thank you very much for your good suggestion. The incubation time and temperature for AR induction were 38.5℃ for 30min. We have replenished the specific incubation time and temperature for AR induction in “Revised manuscript”. Please also see details in Lines 144 in this revision.

Comments 4: Line 114, 127, I suppose the authors would like to express “sperm were wash with PBS by centrifugation (xxx g, xx min)”. What is the reason that the authors used different centrifugation forces and time?

Response 4: We have re-written this part according to the Reviewer’s suggestion. Change lines 129 and 145 to "sperm were wash twice with PBS by centrifugation (400 g, 5 min)" and" after incubation, all samples from two groups were centrifuged (500 g, 5 min) to get deposit (sperm) , sperm were wash thrice with PBS by centrifugation (400 g, 5 min)", respectively.

In fact, Prolonged exposure of sperm to the seminal plasma during liquid in vitro storage creates an unphysiological situation [1,2]. Thus, it is now more routine to separate the sperm from the seminal plasma [3,4]. However, centrifugal on sperm have a certain degree of damage. In mouse [5], human [6-8], dog [9] and stallion [10,11], it has been proven that the centrifugation force and the duration influences both the sperm recovery and yield. Using lower centrifugal forces and shorter centrifugation times leads to lower sperm recovery due to a lack of complete pelleting and, consequently, the loss of unpelleted cells upon supernatant removal [12].In contrast, using higher centrifugal forces and longer centrifugation times leads to greater sperm sedimentation but has a deleterious effect on sperm reducing their motility and viability [13,14], In the process of this experiment, we found that ram semen had the least damage to sperm in the washing process of 400g for 5 minutes, but the sedimentation effect of sperm obtained under this centrifugal force was not enough to support the later test. Therefore, the centrifugal force was slightly increased to minimize the damage of centrifugation to sperm. we finally performed centrifugation in two ways.

References:

[1]. Neila-Montero M, Riesco MF, Alvarez M, et al. Centrifugal force assessment in ram sperm: identifying species-specific impact. Acta Vet Scand. 2021, 63(1), 42.

[2]. Martí E, Pérez-Pé R, Muiño-Blanco T, Cebrián-Pérez JA. Comparative study of four different sperm washing methods using apoptotic markers in ram spermatozoa. J Androl. 2006, 27, 746–53.

[3]. Aitken RJ, Clarkson JS, Clarkson SJ. Significance of reactive oxygen species and antioxidants in defining the efficacy of sperm preparation techniques. J Androl. 1988, 9, 367–76.

[4]. Mortimer D. Sperm recovery techniques to maximize fertilizing capacity. Reprod Fertil Dev. 1994, 6, 25.

[5]. Katkov II, Mazur P. Influence of centrifugation regimes on motility, yield, and cell associations of mouse spermatozoa. J Androl. 1998, 19, 232–41.

[6]. Shekarriz M, DeWire DM, Thomas AJ, Agarwal A. A method of human semen centrifugation to minimize the latrogenic sperm injuries caused by reactive oxygen species. Eur Urol. 1995, 28, 31–5.

[7]. Fredricsson B, Kinnari S. Vitality and morphology of human spermatozoa studies on the resistance to storage and centrifugation and on the removal of dead spermatozoa. Andrologia. 1979, 11, 135–41.

[8]. Mack SR, Zaneveld LJD. Acrosomal enzymes and ultrastructure of unfrozen and cryotreated human spermatozoa. Gamete Res. 1987, 18, 375–83.

[9]. Rijsselaere T, Van Soom A, Maes D, de Kruif A. Effect of centrifugation on in vitro survival of fresh diluted canine spermatozoa. Theriogenology. 2002, 57, 1669–81.

[10]. Knop K, Hoffmann N, Rath D, Sieme H. Effects of cushioned centrifugation technique on sperm recovery and sperm quality in stallions with good and poor semen freezability. Anim Reprod Sci. 2005, 89, 294–7.

[11]. Ferrer MS, Lyle SK, Eilts BE, Eljarrah AH, Paccamonti DL. Factors affecting sperm recovery rates and survival after centrifugation of equine semen. Theriogenology. 2012, 78, 1814–23.

[12]. Carvajal G, Cuello C, Ruiz M, Vázquez JM, Martínez EA, Roca J. Effects of centrifugation before freezing on boar sperm cryosurvival. J Androl. 2004, 25, 389–96.

[13]. Hoogewijs M, Rijsselaere T, De Vliegher S, Vanhaesebrouck E, De Schauwer C, Govaere J, et al. Influence of different centrifugation protocols on equine semen preservation. Theriogenology. 2010, 74, 118–26.

[14]. Cardullo RA, Cone RA. Mechanical immobilization of rat sperm does not change their oxygen consumption rate. Biol Reprod. 1986, 34, 820–30.

Comments 5: Line 139-140, it is necessary to add the dilution rate of those antibodies used.

Response 5: Thank you very much for your good suggestion. We had added the dilution rate of those antibodies used in line139-140 according to the Reviewer’s suggestion: anti-Tyr-P (CST9411, 1:20, MA, USA) and anti-α-tubulin (Proteintech66031-1-Ig, 1:300, Wuhan, China). Please also see details in Line 160-161.

Comments 6: Line 142, instead of 30 mg frozen sperm pellet, I would prefer to indicate the total sperm number for each sample because that will be better controlled and easy to be duplicated. Furthermore, when the comparison of proteins profiles between samples before and after capacitation, it is reasonable to compare that among the samples of same sperm number.

Response 6: Thank you very much for your good suggestion. We had replenished the total sperm number for each sample in “Revised manuscript”. For comparison of protein profiles between samples before and after capacitation, we used samples with the same number of sperm for comparison. After incubation, sperm samples were centrifuged, washed, and collected to approximately 1 x 107 AR spermatozoa before protein concentration was determined by the BCA kit. Finally, each protein sample (30 mg) was subjected to enzymolysis. Please also see details in Line 147, Line 163 and Line 173.

Comments 7: Line 291-292 were from the template and should be deleted.

Response 7: According to your suggestions we have deleted this part.

Reviewer 4 Report

Comments and Suggestions for Authors

Congratulations, I really enjoyed reading it

I just have a few observations:

- line 291: why you put this were?

I don't think there's any need to put a subtitle with this information. If you integrate the information into the text, I think it would be strange to put a subtitle with just this

please review this. 

Author Response

Comments 1: line 291: why you put this were? I don't think there's any need to put a subtitle with this information. If you integrate the information into the text, I think it would be strange to put a subtitle with just this.

Response: Thank you for your valuable suggestion on our manuscript. It’s very helpful for improving our manuscript. In fact, line 291-292 were from the template and we have deleted this part.
